# Auto-Classification of Parkinson’s Disease with Different Motor Subtypes Using Arterial Spin Labelling MRI Based on Machine Learning

**DOI:** 10.3390/brainsci13111524

**Published:** 2023-10-29

**Authors:** Jinhua Xiong, Haiyan Zhu, Xuhang Li, Shangci Hao, Yueyi Zhang, Zijian Wang, Qian Xi

**Affiliations:** 1Department of Radiology, Shanghai East Hospital, Tongji University School of Medicine, No. 150 Jimo Road, Pudong New Area, Shanghai 200120, China; xjh941116@sina.com (J.X.);; 2Department of Radiology, Shanghai Tongji Hospital, Tongji University School of Medicine, No. 389 Xincun Road, Putuo District, Shanghai 200065, China; 3School of Computer Science and Technology, Donghua University, No. 2999 North Renmin Road, Songjiang Area, Shanghai 200000, China

**Keywords:** Parkinson’s disease, motor subtypes, arterial spin labelling, machine learning, support vector machine

## Abstract

The purpose of this study was to automatically classify different motor subtypes of Parkinson’s disease (PD) on arterial spin labelling magnetic resonance imaging (ASL-MRI) data using support vector machine (SVM). This study included 38 subjects: 21 PD patients and 17 normal controls (NCs). Based on the Unified Parkinson’s Disease Rating Scale (UPDRS) subscores, patients were divided into the tremor-dominant (TD) subtype and the postural instability gait difficulty (PIGD) subtype. The subjects were in a resting state during the acquisition of ASL-MRI data. The automated anatomical atlas 3 (AAL3) template was registered to obtain an ASL image of the same size and shape. We obtained the voxel values of 170 brain regions by considering the location coordinates of these regions and then normalized the data. The length of the feature vector depended on the number of voxel values in each brain region. Three binary classification models were utilized for classifying subjects’ data, and we applied SVM to classify voxels in the brain regions. The left subgenual anterior cingulate cortex (ACC_sub_L) was clearly distinguished in both NCs and PD patients using SVM, and we obtained satisfactory diagnostic rates (accuracy = 92.31%, specificity = 96.97%, sensitivity = 84.21%, and AUCmax = 0.9585). For the right supramarginal gyrus (SupraMarginal_R), SVM distinguished the TD group from the other groups with satisfactory diagnostic rates (accuracy = 84.21%, sensitivity = 63.64%, specificity = 92.59%, and AUCmax = 0.9192). For the right intralaminar of thalamus (Thal_IL_R), SVM distinguished the PIGD group from the other groups with satisfactory diagnostic rates (accuracy = 89.47%, sensitivity = 70.00%, specificity = 6.43%, and AUCmax = 0.9464). These results are consistent with the changes in blood perfusion related to PD subtypes. In addition, the sensitive brain regions of the TD group and PIGD group involve the brain regions where the cerebellothalamocortical (CTC) and the striatal thalamocortical (STC) loops are located. Therefore, it is suggested that the blood perfusion patterns of the two loops may be different. These characteristic brain regions could become potential imaging markers of cerebral blood flow to distinguish TD from PIGD. Meanwhile, our findings provide an imaging basis for personalised treatment, thereby optimising clinical diagnostic and treatment approaches.

## 1. Introduction

Parkinson’s disease (PD) is a common neurodegenerative disorder that becomes increasingly prevalent with age [1]. The typical symptoms of PD are rigidity, bradykinesia, tremor, and postural instability [2], which are caused by a profound loss of dopaminergic neurons from the basal ganglia [3]. Additionally, many environmental and genetic factors exert an influence on the risk of PD, with different factors predominating in different patients. These factors converge on specific pathways, including mitochondrial dysfunction, oxidative stress, protein aggregation, impaired autophagy, and neuroinflammation [4]. Several pathophysiological concepts, pathways, and mechanisms, including the presumed roles of α-synuclein misfolding and aggregation, Lewy bodies, oxidative stress, iron and melanin, deficient autophagy processes, insulin and incretin signalling, T-cell autoimmunity, the gut–brain axis, and the evidence that microbial (viral) agents, may induce molecular hallmarks of neurodegeneration [5]. The Unified Parkinson’s Disease Rating Scale (UPDRS) is the most commonly used scale to assess the motor symptoms of PD patients [6]. Based on the UPDRS score, Jankovic was the first person who proposed classifying idiopathic PD into tremor-dominant (TD) and postural instability and gait difficulty (PIGD) subtypes [7]. PD patients with different motor subtypes have different disease progression and prognoses. The PIGD subtype has more severe motor and cognitive impairment and worse response to drug treatment than the TD subtype [7,8]. Therefore, it is of great significance to improve the clinical classification of PD for individualised treatment. Meanwhile, there is an increasing interest in the analysis of variability in clinical presentation, which reflects the existence of multiple subtypes of, and heterogeneous progression in, PD. The identification of patient subgroups within PD has significant implications for generating hypotheses on defining the heterogeneity of PD, understanding etiopathogenic mechanisms, and developing treatments [9].

At present, the classification of PD subtypes is mainly based on the UPDRS. The brain imaging neural markers of PD are far from reaching a consensus. Excavating the neural mechanism under the imaging is conducive to promoting the differential diagnosis of PD subtypes so as to optimise the clinical diagnosis and treatment. In recent years, neuroimaging studies have shown that cerebrovascular lesions are common in PD patients. Therefore, PD is considered a disease related to abnormal cerebrovascular function [10]. This issue was also described in the context of atypical parkinsonisms, such as corticobasal syndrome (CBS), characterised by both motor and higher cortical dysfunctions. Furthermore, ischemia is the primary risk factor for vascular CBS. Cerebral hypoperfusion can play a significant role in neuropathological changes in neurodegenerative diseases, CBS included [11]. Previous studies have confirmed that dopaminergic neurons are attached to brain microvessels and cerebral blood flow (CBF) changes due to metabolic reduction caused by neuronal degeneration and death [12]. However, this basic pathological change reflected in cerebral blood perfusion in patients with different motor subtypes of PD has not been confirmed by definite studies.

Arterial spin labelling (ASL) is a magnetic resonance imaging (MRI) perfusion technique that enables the quantification of CBF without the use of intravenous gadolinium contrast [13]. Regional CBF measured by ASL is relatively stable and is considered to reflect the functional activity of the brain directly [14]. Studies have shown that ASL technology can detect signs of neurodegeneration at an earlier stage and can be used to monitor changes in CBF during the progression of the disease [15]. There was no difference in whole-brain CBF in TD patients compared to PIGD patients. The prolonged arterial arrival time appeared more diffuse in the TD group than in the PIGD group. The PIGD group had a more predominantly posterior pattern of hypoperfusion and, indeed, basal ganglia hyperperfusion than the more temporo-parieto-frontal hypoperfusion of the TD group (which did not show areas of hyperperfusion) [16]. To our knowledge, the PD subtypes differences revealed in specific brain regions of CBF have not been previously investigated. Currently, resting-state functional MRI is widely utilised in the study of PD motor subtypes [17,18,19,20], while there are few studies on ASL-MRI. It is necessary to consider an imaging marker of CBF to distinguish TD from PIGD.

With the development of machine learning technology, support vector machine (SVM) has been widely used in the early diagnosis and classification of PD due to its excellent performance [21,22,23,24]. SVM aims to find the maximum interval between classes, which is known as the optimal decision boundary. This helps enhance the generalisation performance of classification, which is particularly important for medical data classification, where accuracy is paramount. SVM has shown excellent performance for the classification of PD motor subtypes using neuroimaging data or 3D kinematic data [25,26]. The aim of this study was to use SVM to perform automatic classification on ASL-MRI data and explore the neuroimaging markers of PD subtypes in cerebral blood perfusion.

## 2. Materials and Methods 

### 2.1. Subjects

We enrolled 38 subjects in this study, including 17 normal controls (NCs). Based on the Unified Parkinson’s Disease Rating Scale (UPDRS) subscores, there were 11 TD and 10 PIGD patients among the 21 PD patients. This study was reviewed and approved by the Ethics Review Committee of Shanghai East Hospital. Written informed consent was obtained from all subjects. All subjects underwent by the following tests: (1) Mini-Mental State Examination (MMSE); (2) Modified Hoehn and Yahr clinical grading scale; (3) Movement Disorder Society-Sponsored Revision UPDRS.

The inclusion criteria of PD patients were as follows: (1) age 50–75 years old, tremor was the main symptom, Hoehn and Yahr stages II–IV; (2) clear and effective treatment with dopaminergic drugs; (3) no other systemic malignant tumours. The inclusion criteria for the NCs were as follows: (1) sex and age matching those of participants in the PD group (there was no statistically significant difference (*p* > 0.05)); (2) the patients were healthy without nervous system diseases. The exclusion criteria were as follows: (1) history of recurrent stroke, transient ischemic attack, brain injury, and encephalitis; (2) symptoms during the use of antipsychotic drugs; (3) serious heart, liver, and kidney diseases and mental disorders; (4) severe autonomic nervous dysfunction occurring in the early stage of the disease; (5) inability to cooperate with the examination due to various reasons (such as illiteracy, advanced age, hearing impairment, claustrophobia, etc.).

### 2.2. Magnetic Resonance Imaging

All subjects were scanned with a M750w 3.0T GE Signa MRI system (GE Healthcare, Chicago, IL, USA) equipped with a 32-channel phased-array head coil. The subjects were in a resting state during the acquisition of ASL-MRI data. During the scanning, the subjects were in the supine position with their head fixed using a fixed band, and earplugs were placed in both ears to reduce scanner noise. All subjects were asked to limit their head movements as much as possible. The three groups of subjects were scanned with the same sequence under the same parameters. The sequences included conventional MRI sequences (T1WI and T2WI), DWI, and ASL sequences. Other nervous system lesions, such as multiple cerebral infarctions, hydrocephalus, and intracranial tumours, can be excluded by conventional MRI scans in selected subjects.

Conventional MRI scans were performed, including cross-sectional T1WI (repetition time (TR) = 2000 ms, echo time (TE) = 20 ms, field of view (FOV) = 250 × 221 mm, matrix = 400 × 250, and slice thickness/slice distance = 7 mm/0.6 mm); T2WI (TR = 3000 ms, TE = 80 ms, FOV = 250 × 221 mm, matrix = 436 × 295, and slice thickness/slice distance = 7 mm/0.6 mm); FLAIR (TR = 11,000 ms, TE = 120 ms, FOV = 250 × 221 mm, matrix =240 × 160, and slice thickness/pitch = 7 mm/0.6 mm); and DWI (TR = 2634 ms, TE = 58 ms, FOV = 230 × 230 mm, matrix = 140 × 136, and slice thickness/slice distance = 6 mm/0.6 mm). The ASL scanning parameters were as follows: TR = 4854 ms; TE = 10.7 ms; post-labelling delay time = 2025 ms; spiral arm = 8; sampling point = 512; flip angle (FA) = 111°; FOV = 240 mm × 240 mm; reconstruction matrix = 128 × 128; slice thickness = 4 mm, no septum; slice number = 36, axial position; number of excitations (NEX) = 3. The intraslice resolution was 1.9 mm × 1.9 mm, and the scan time was 282 s.

### 2.3. Statistical Analysis

We analysed the general information and clinical scale data of the subjects using the Statistical Package for the Social Sciences version 26.0 (IBM Corp., Armonk, NY, USA) and compared the gender distribution of the groups by performing a chi-square test. For quantitative data, we first performed a normality test (Shapiro–Wilk test) and homogeneity of variance test. We expressed normally distributed data as the means ± standard deviations. We compared the three groups using one-way analysis of variance and compared pairs of groups using a two-sample *t*-test. We expressed non-normally distributed data as M (P25, P75). The Kruskal–Wallis test was used for comparisons among the three groups, and a two-sample nonparametric *t*-test was used for comparisons between the two groups. *p* < 0.05 was considered statistically significant.

### 2.4. Data Preprocessing

#### 2.4.1. Feature Extraction

Due to the long time required for MR image acquisition, it is difficult for PD patients with a tremor to avoid head movement, which can affect the subsequent data analysis. Therefore, the brain images in the first 10 time points of each subject were discarded to ensure the stability of the data signals. The brain images in the remaining time series were corrected by interlayer time correction, strict head movement correction, brain normalization, and image smoothing using the SPM spatial template to minimise the possible influence of head movement.

After correction, the automated anatomical atlas 3 (AAL3) brain region template (including 170 brain regions) was selected. We used SPM to register the AAL3_1 mm template with the ASL image to obtain images with the same size and shape. According to the brain regions defined by AAL3, we obtained the location coordinates contained by each brain region in the AAL3 image and then obtained the voxel values of the corresponding brain region location of the subject. A csv file was generated for each brain region containing the voxel values of that region for all subjects, so we obtained the voxel values for 170 brain regions. Due to the different size of each brain region, the number of corresponding voxel values also varied. Before the data were entered into the SVM classifier, we only normalised the voxel data of the current brain region without changing the data size. Consequently, the length of the feature vector depended on the number of voxel values in each brain region. The length of these feature vectors varied from brain region to brain region. However, the length of the feature vectors was consistent for each brain region. For example, in Acc_pre_L, the number of voxels for per patient was 626, while, in Angular_R, the number of voxels was 1751.

#### 2.4.2. Model Classification and Validation

Patients were classified according to the obtained voxel values in each brain region. The data of NC, PIGD patients, and TD patients were referred to the classification methods of previous similar studies [27], and three binary classification models were proposed: “NC vs. others”, “PIGD vs. others”, and “TD vs. others”.

The data were normalised and used to construct SVM classifiers based on the Sklearn library. We adopted the leave-one-out cross-validation (LOOCV) method to estimate the performance of the classifiers. Given a set of data samples, the classifier removed one data sample in each trial, and the classifier was trained on the remaining data samples. The removed samples were used for model testing [28]. Since the feature vector size of each brain region is different, we trained the model for the same brain region of all subjects in each experiment to test the diagnostic effect under the current brain region. According to the classification standard of the AAL3 brain region template, a total of 170 brain regions were shown. Therefore, experiments were conducted for all 170 brain regions. In the experiment, each subject’s current brain region would be used as the test set in turns due to adopting the LOOCV. For example, when we targeted the Thal_VA_L in the AAL3 template for the experiment, the voxel value of Thal_VA_L for each subject was taken as a sample. When we performed experiments on other brain regions, since the number of voxel values in each brain region was different, the length of the feature vector in each experiment depended on the number of voxel values in each brain region. The model performance indicators of accuracy, sensitivity, specificity, and maximum area under the curve (AUCmax) in the receiver operating characteristic (ROC) analysis were used to evaluate the classification performance of the SVM model. The overall procedure of data preprocessing, feature extraction, model classification, and validation is displayed in Figure 1. 

## 3. Results

### 3.1. Demographic and Clinical Study

There was no significant difference in age, sex, or disease duration among the three groups (*p* > 0.05, Table 1). There was no significant difference in UPDRS score or H&Y grade between the TD group and PIGD group (*p* > 0.05, Table 1). There was no significant difference in MMSE scores between the TD group and the PIGD group (*p* > 0.05, Table 1).

### 3.2. Classifier Performance Assessment

After SVM screening, a total of 4 brain regions with high accuracy were selected from 170 brain regions in the AAL3 template. According to the performance analysis of three binary classification models, we found that the left subgenual anterior cingulate cortex (ACC_sub_L) of the NCs was more sensitive to classification than that of the PD patients. The proposed classifier differentiated PD patients and NCs with diagnostic accuracy, sensitivity, and specificity of 81.58%, 76.47%, and 85.71%, respectively. At the same time, the ROC analysis showed that the AUCmax reached 0.8992.

For the right supramarginal gyrus (SupraMarginal_R), SVM distinguished the TD group from the other groups with diagnostic accuracy, sensitivity, and specificity of 84.21%, 63.64%, and 92.59%, respectively, and the AUC value was 0.9192. For the right intralaminar of the thalamus (Thal_IL_R), SVM could distinguish the PIGD group from the other groups with a diagnostic accuracy, sensitivity, and specificity of 89.47%, 70.00%, and 96.43%, respectively, and the AUC value was 0.9464. For the left lateral geniculate of the thalamus (Thal_LGN_L), the accuracy of the TD and PIGD classification was above 75%, but the AUC value was relatively low (Table 2 and Figure 2).

### 3.3. Visualisation of the Most Sensitive Features

According to the sensitive brain regions screened by SVM, we input four related sensitive brain regions into BrainNetViewer for visualisation [29] and displayed the related brain regions intuitively (Figure 3).

## 4. Discussion 

This study introduces a SVM-based classifier for the differential diagnosis of PD patients with different motor subtypes using ASL-MRI data for the first time. In general, the proposed classifier has high classification performance in the four brain regions, showing a satisfactory classification ability. The diagnostic accuracy, sensitivity, specificity, and AUCmax value are high, which are almost consistent with the evaluation of the clinical scales. This indicates the feasibility of using ASL-MRI data for the automatic classification of PD subtypes. We also find that the voxel values of the four related brain regions are the most sensitive classification features, which can be used as potential neuroimaging markers for PD subtypes in cerebral blood perfusion.

The AAL3 brain template used in this study helped to further divide the brain regions into detailed subregions [30]. These regions are of interest in many neuroimaging studies and studies of psychiatric and neurological disorders [31,32,33,34]. Compared to radiomics features extracted from ROIs (left and right caudate and putamen) in MRI images and DAT SPECT images [35], we paid more attention to the extraction and selection of the features of the whole brain. Numerous new data-driven methods, such as biclustering or triclustering, seem to have been proposed for subtyping from neuroimaging data [36]. Unlike data-driven methods applied to schizophrenia research [37,38], SVM has gained significant popularity for the early diagnosis and classification of PD. The SVM algorithm in machine learning was used for the classification model. We utilised a linear kernel SVM, which is also a linear classifier. This classifier has demonstrated exceptional classification performance and interpretability, rendering it extensively utilised in various research endeavours. In our data sample, the number of subjects in the control group was large, while the number of subjects in the other categories was relatively small, and the categories were unbalanced. SVM can handle unbalanced data by adjusting the regularisation parameter C to ensure that the model is not biased toward the dominant category. In the conducted experiment, a range of regularisation parameter C values from 1 to 1000 were explored. Based on the obtained experimental results, it was determined that the current value of C exhibited optimal efficacy. Compared to deep learning and random forest, SVM is more suitable for the research of small samples. Due to the existence of a “black box”, deep learning is not as interpretable as SVM. LOOCV was used as the validation method instead of k-fold cross-validation, because it is suitable for small sample studies. Future research could explore the application of unsupervised machine learning for the data-driven identification of motor subtypes in PD. Previous studies have employed clustering methods such as unsupervised hierarchical clustering, KMeans, and random forest clustering to identify subtypes of PD [35,39,40].

Voxel-based morphometry (VBM) is a neuroimaging technique that investigates focal differences in brain anatomy [41]. Furthermore, VBM is widely used for neurodegenerative and psychiatric diseases [41,42,43,44,45]. In previous studies using VBM to distinguish idiopathic PD patients from normal subjects, the analysis was based on multiple machine learning classifiers. The results indicated that the logistic method and support vector machine showed the best performance [46]. However, a possible problem with these approaches is that the evaluated regions are not the most relevant to the pathogenesis of PD. We found that SVM could distinguish the NC group from the PD group in ACC_sub_L, which was consistent with a previous study on the cingulate cortex in PD. Evidence has been provided for a new conceptualisation of the connectivity and functions of the cingulate cortex in emotion, action, and memory [47]. In addition, VBM has been used in many studies of mild cognitive impairment in PD to show reduced thickness in the anterior cingulate cortex and posterior cingulate cortex. Regional CBF is altered in association with the verbal intelligence quotient in the posterior cingulate cortex and anterior midcingulate cortex and in association with executive impairments in the anterior cingulate cortex [48].

In particular, a structural MRI study showed decreased cerebellar grey matter and increased Sulc (a measure of sulcal depth) in the right supramarginal gyrus in the TD subtype [49,50]. SupraMarginal_R has been shown to play an important role in various cognitive functions [51]. The intralaminar nuclei, through extensive projections to the striatum and cortex, participates in a range of behaviours, including sensorimotor coordination, pain modulation, arousal, and cognition [52]. In general, PIGD subtypes mainly involve changes in the basal ganglia output-related circuitry (striatal thalamocortical loop, STC loop), while TD subtypes involve an additional downstream compensation mechanism consisting of the cerebellothalamocortical (CTC) loop [53]. Based on the more than 30 quantitative PD studies performed to date, it seems safe to conclude that the resting state in PD patients is characterised by various degrees of hypoperfusion and hypometabolism in cerebral cortical structures (mostly frontoparietal) and possibly also in certain subcortical structures [54]. A recent study using ASL revealed that TD exhibited more hypoperfusion in the temporo–parieto–frontal network, while PIGD showed hypoperfusion in a predominantly posterior pattern, as well as hyperperfusion in the basal ganglia [55]. The TD group showed a higher classification performance in SupraMarginal_R, while the PIGD group showed the highest classification performance in Thal_IL_R. This is consistent with the changes in blood perfusion related to the PD subtype. The sensitive brain regions of the TD group and PIGD group were in the brain regions involved in the CTC and STC loops, so it is suggested that the blood perfusion patterns of the two pathways may be different.

This study still had several limitations. First, the sample size included in this study was small. Additionally, the relatively wide age range of participants may have had an impact on the results due to the small sample. Therefore, the conclusions of this study need to be further verified by large-sample and multicentre data. Second, the detection results of cerebral blood flow perfusion by ASL are easily affected by the post-labelling delay (PLD) time, slice thickness, matrix, and other parameters; thus, people of different ages need different PLD times. The 2025 ms PLD time interval selected in this study conforms to the requirements of the 2014 expert consensus for a single PLD time of pseudo-continuous arterial spin labelling to minimise its influence in most adults [56]. Third, the reliability of the classification model would be further improved if a multimodal comparative study were to be carried out by combining biological markers; other modalities such as MRI, PET, or SPECT; and other imaging methods. In the future, on the basis of expanding the sample size and integrating other modality images, we further will optimise the algorithm for quantitative research to enhance the accuracy of the sensitive features and provide multidimensional neuroimaging markers for clinical diagnosis and treatment.

## 5. Conclusions

In conclusion, we introduced a classification method based on machine learning to classify ASL-MRI images of PD patients with different motor subtypes and found that the classification efficiency was high in four brain regions. In addition, ACC_sub_L can be used as a neuroimaging marker for the classification of PD and NCs. SupraMarginal_R and Thal_IL_R are within the range of the CTC and STC loops, which is helpful for investigating the cerebral blood perfusion patterns of the two loops. These characteristic brain regions could become potential imaging markers of CBF to distinguish TD from PIGD. It can help to explain the differences in the anatomical and clinical symptoms of different PD motor subtypes and provide an imaging basis for research on the neuropathological mechanism and personalised treatment, thereby optimising clinical diagnostic and treatment approaches.

## Figures and Tables

**Figure 1 brainsci-13-01524-f001:**
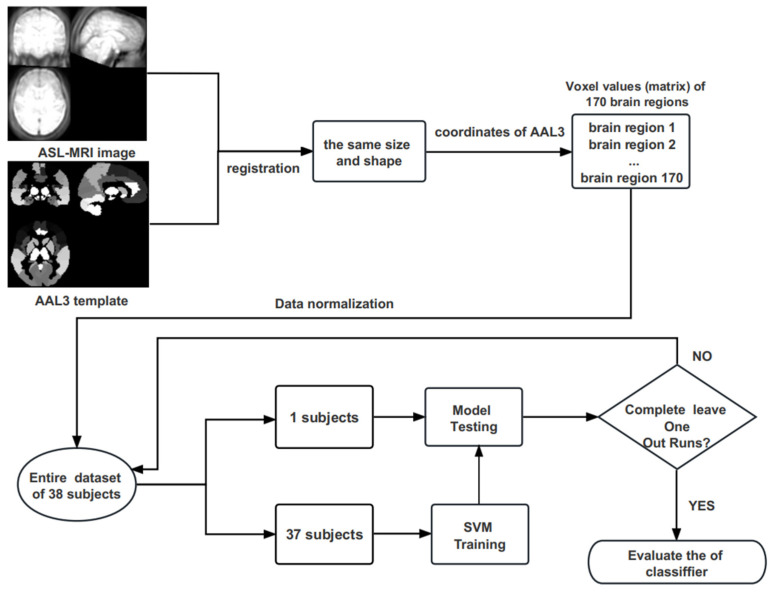
The flowchart of data preprocessing, feature extraction, model classification, and validation.

**Figure 2 brainsci-13-01524-f002:**
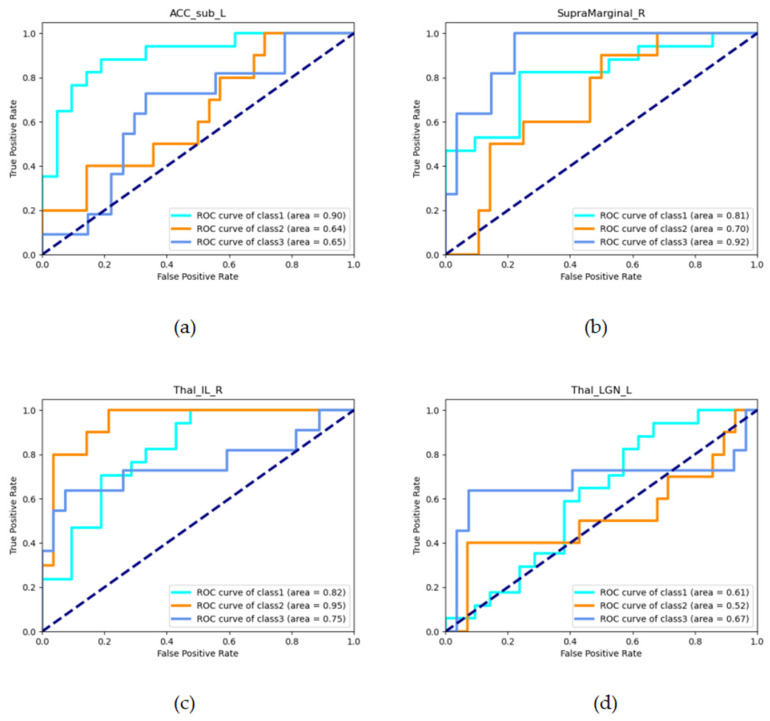
The ROC curve of the SVM classifier for sensitive brain regions. (**a**) ROC of ACC_sub_L in three binary classification models; (**b**) ROC of SupraMarginal_R in three binary classification models; (**c**) ROC of Thal_IL_R in three binary classification models; (**d**) ROC of Thal_LGN_L in three binary classification models; Class 1 = “NC vs. others”, Class 2 = “PIGD vs. others”, and Class 3 = “TD vs. others”.

**Figure 3 brainsci-13-01524-f003:**
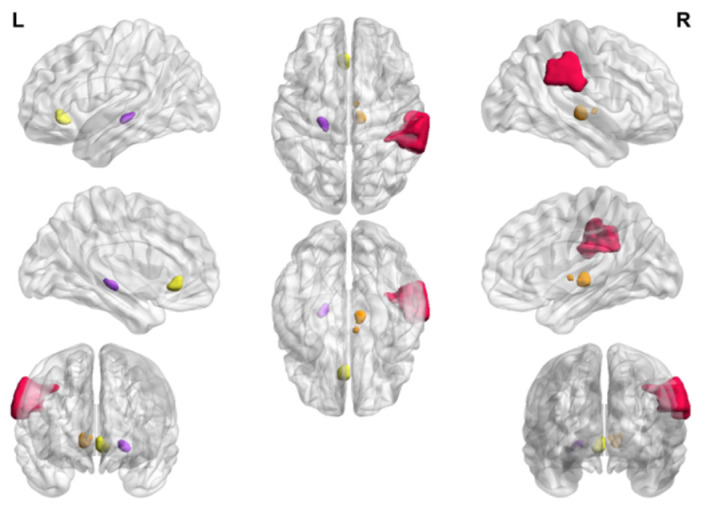
Visualisation of the relevant brain regions. Red region = SupraMarginal_R; orange region = Thal_IL_R, yellow region = ACC_sub_L, and purple region = Thal_LGN_L.

**Table 1 brainsci-13-01524-t001:** Comparison of general clinical data among the three groups.

Groups	NC	TD	PIGD	*p*-Value
Number of subjects	17	11	10	-
Age (year)	64 (52~68)	68 (55~70)	69.500 (67.25~68.75)	0.052
Sex (M/F)	5/12	5/6	4/6	0.671
Disease duration (year)	-	6 (4~6)	4/6	0.152
H&Y	-	1.500 (1~2)	1.500 (1.375~2)	0.809
UPDRS	-	36.730 ± 15.021	39.700 ± 11.870	0.623
MMSE	-	27.820 ± 3.682	26.000 ± 3.582	0.260

H&Y: Hoehn and Yahr stage; UPDRS: Unified Parkinson’s Disease Rating Scale; MMSE: Mini-Mental State Examination; NC: normal control; TD: tremor-dominant; PIGD: postural instability and gait difficulty.

**Table 2 brainsci-13-01524-t002:** The diagnostic performance of sensitive brain regions for the three binary classifications.

Brain Regions	Groups	Accuracy	Sensitivity	Specificity	AUC
	NC vs. others	81.58%	76.47%	85.71%	89.92%
ACC_sub_L	PIGD vs. others	65.79%	40.00%	75.00%	63.57%
	TD vs. others	68.42%	54.55%	74.07%	64.98%
SupraMarginal_R	NC vs. others	76.32%	76.47%	76.19%	80.67%
	PIGD vs. others	73.68%	50.00%	82.14%	70.00%
	TD vs. others	84.21%	63.64%	92.59%	91.92%
	NC vs. others	73.68%	70.59%	76.19%	81.79%
Thal_IL_R	PIGD vs. others	89.47%	70.00%	96.43%	94.64%
	TD vs. others	81.58%	63.64%	88.89%	75.42%
	NC vs. others	57.89%	64.71%	52.38%	60.50%
Thal_LGN_L	PIGD vs. others	76.32%	30.00%	92.86%	52.14%
	TD vs. others	78.95%	45.45%	92.59%	67.34%

ACC_sub_L: the left subgenual of anterior cingulate cortex; SupraMarginal_R: the right supramarginal gyrus; Thal_IL_R: the right intralaminar of the thalamus; Thal_LGN_L: the left lateral geniculate of the thalamus; NC: normal control; TD: tremor-dominant; PIGD: postural instability and gait difficulty.

## Data Availability

All data reported in this manuscript will be made available from the corresponding author upon reasonable request.

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
