# Peer review of "Auto-Classification of Parkinson’s Disease with Different Motor Subtypes Using Arterial Spin Labelling MRI Based on Machine Learning"

_brainsci, 2023, doi:10.3390/brainsci13111524_

Round 1

Reviewer 1 Report

Comments and Suggestions for Authors

Comments and Suggestions for Authors

1)Were the participants in resting state or task during acquisition of ASL-MRI data..This should be included in abstract and Section 2.2 (Magnetic Resonance Imaging).Faetures extracted and parameters of SVM also needs to be included in Abstract

2)The authors have missed out discussing some parkinsons articles related to subtyping in introduction…No articles from 2023?.

3)The authors should include in introduction a discussion about relevant ML subtyping studies in Parkinsons using neuroimaging data… No articles from 2023

4)What is the motivation for subtyping using ASL-MRI data seems to be missing in introduction. Also ASL-MRI based subtypes has already been done?

5)Why was AAL3 brain region template selected…Any reference?

6) Numerous new data driven methods seem to have proposed for subtyping from neuroimaging data.. Did the authors consider them.. Write about these in methodology?

e.g.:

https://pubmed.ncbi.nlm.nih.gov/30946659/

https://bmcbioinformatics.biomedcentral.com/articles/10.1186/s12859-022-04733-8

https://pubmed.ncbi.nlm.nih.gov/34049447/

7)Authors say “Patients were classified in each brain region based on the obtained voxel values of 144 brain regions”…what are actually the features? size of feature vector?

8) What is the motivation for selecting an SVM classifier? For a binary classification cant we use a simple linear classifier? More info on length of feature vector for each subject

9)Perhaps the authors could consider uploading the data and software code to make things easy for reviewer.

10)Discussion should be rewritten after reading all articles…Comparison of obtained results should be done with previous ASL-MRI based subtyping of parkinsons..A paragraph highlighting discussions about results obtained here using SVM could be compared with other Machine learning classifier(Random Forest,Bayes,etc) studies using neuroimaging data

11)Fig 1 and 3 could be bettered.

12)DOI for numerous references missing..Inculde them

Comments on the Quality of English Language

The authors are requested to improve quality of english language

Author Response

Thank you very much for taking the time to review this manuscript. Please find the detailed responses below and the corresponding revisions in the re-submitted files.

Response 1: Thank you for pointing this out. We agree with this comment. The participants were in resting state during the acquisition of ASL-MRI data. This has been included in abstract and Section 2.2 (Magnetic Resonance Imaging). Faetures extracted and parameters of SVM have also bee included in Abstract.

Response 2: We agree with this comment. We have discussed some parkinsons articles related to subtyping in introduction. Mention exactly where in the revised manuscript this change can be found–page 2, paragraph 3, and line 10.

Response 3: In introduction, We have a discussion about relevant ML subtyping studies in Parkinsons using neuroimaging data. Mention exactly where in the revised manuscript this change can be found–page 3, paragraph 1, and line 6.

Response 4:Currently, Resting-state functional MRI is widely utilized in the study of PD motor subtypes, while there are few studies on ASL-MRI. It is necessary to consider an imaging marker of CBF to distinguish TD. Mention exactly where in the revised manuscript this change can be found–page 2, paragraph 3 , and line 10. The Unified Parkinson's Disease Rating Scale (UPDRS) is the most commonly used scale to assess the motor symptoms of PD patients. This study introduces an SVM-based classifier for the differential diagnosis of PD patients with different motor subtypes using ASL-MRI data for the first time. Meanwhile , Our findings provide an imaging basis for personalized treatment, thereby optimizing clinical diagnostic and treatment approaches. Mention exactly where in the revised manuscript this change can be found in Abstract.

Response 5: The AAL3 brain template used in this study helped to further divide the brain regions into detailed subregions. These regions are of interest in many neuroimaging studies and studies of psychiatric and neurological disorders. Mention exactly where in the revised manuscript this change can be found–page 8, paragraph 2 , and line 1.

Response 6:Numerous new data driven methods, such as biclustering or tri-clusterin, seem to have proposed for subtyping from neuroimaging data. Unlike data driven methods applied to schizophrenia research, SVM has gained significant popularity for the early diagnosis and classification of PD. Future research could explore the application of unsupervised machine learning for data-driven identification of motor subtypes in PD. Mention exactly where in the revised manuscript this change can be found–page 8, paragraph 2 , and line 6.

Response 7: Due to the different size of each brain region, the number of corresponding voxel values also varies. Before the data were entered into the SVM classifier, we only normalized the voxel data of the the current brain region without changing the data size. Consequently, the length of the feature vector depends on the number of voxel values in each brain region. The length of these feature vectors varies from brain region to brain region, but for each brain region, the length of the feature vectors is consistent. Mention exactly where in the revised manuscript this change can be found–page 4, paragraph 4 , and line 8.

Response 8: We utilized a linear kernel SVM, which is also a linear classifier. This classifier has demonstrated exceptional classification performance and interpretability, rendering it extensively utilized in various research endeavors. In our data sample, the number of patients in the control group was large, while the number of patients in the other categories was relatively small and the categories were unbalanced. SVM can handle unbalanced data by adjusting the regularization parameter C to ensure that the model is not biased toward the dominant category. Mention exactly where in the revised manuscript this change can be found–page 8, paragraph 2 , and line 11.

Response 9: All data reported in this manuscript will be made available from the corresponding author on reasonable request.

Response 10: Compared with the deep learning and random forest , SVM is more suitable for the research of small samples. Future research could explore the application of unsupervised machine learning for data-driven identification of motor subtypes in PD. Mention exactly where in the revised manuscript this change can be found–page 8, paragraph 2 , and line 17.

Response 11:Fig 1 and 2 have been improved, and explanatory notes have been added in Fig 2 and 3.

Response 12: DOI for numerous references has been added.

Response to Comments on the Quality of English Language

As for the English revisions, we have used editing services and improved the level of English in the process of revision.

Reviewer 2 Report

Comments and Suggestions for Authors

The purpose of this study was to automatically classify different motor subtypes of Parkinson's disease (PD) in arterial spin labeling magnetic resonance imaging (ASL-MRI) data using a support vector machine (SVM). Twenty-one PD patients and 17 normal controls (NC) were recruited and patients were divided into tremor dominant (TD) subtype and postural instability and gait difficulty (PIGD) subtype.

The article raises an interesting question, since the classification of Parkinson's users can facilitate their treatment.

We recommend adding this citation:DOI:10.3233/JPD-202067

On the other hand, the sample is small, as already noted in the conclusions.

At the conclusion of the summary, remove "speculate" and replace it with "suggest" or another word.

The statistics are correct, and so are the images. I have a doubt if an inferential table can be made between the Parkison types.

Author Response

Thank you very much for taking the time to review this manuscript. Please find the detailed responses below and the corresponding revisions in the re-submitted files.

Response 1: Thank you for pointing this out. We agree with this comment. Therefore, We have added this reference:DOI:10.3233/JPD-202067.

Response 2: At the conclusion of the summary, we have removed "speculate" and replace it with "suggest".

Response 3: The model performance indicators of accuracy, sensitivity, specificity and maximum area under the curve (AUCmax) in the receiver operating characteristic (ROC) analysis were used to evaluate the classification performance of the SVM model. Deep learning may show the model architecture and turns the data step by step into the final classification value. This study is based on SVM in machine learning, which is different from the classification method of deep learning.

Reviewer 3 Report

Comments and Suggestions for Authors

The study by Xiong elaborates on the significance of arterial spin labelling MRI based on machine learning in classifying the motor phenotypes of Parkinson's disease (PD). The work, though attempting to indicate vital aspects of the second most common neurodegenerative disease, should be further improved in the following points:

1. In the introduction PD should be presented also in the context of various hypotheses regarding its pathogenesis.

2. Authors indicate the link between PD and cerebrovascular function, it would be valuable to acknowledge links between vascular dysfunction and  parkinsonisms more generally, as this issue was also described in the context of atypical parkinsonisms e.g. Corticobasal Syndrome - Ref.

The Significance of Vascular Pathogenesis in the Examination of Corticobasal Syndrome. Front Aging Neurosci. 2021 May 4;13:668614. doi: 10.3389/fnagi.2021.668614. PMID: 34017244; PMCID: PMC8129188.

3. In the material section of the manuscript authors should state whether transient ischemic attack in the past was also an exclusion criterium.

4. The regions indicated in the classifier performance assessment should be discussed in the context of their possible association with clinical manifestation

5. In the limitation section authors should also indicate the relatively wide age range, which could impact the results due to the small sample.

6. In the discussion authors indicate possibly combined biological markers as PET and SPECT, however the issue of advantages and disadvantages of these methods in the comparison with the tool assessed in this study is not brought up in the earlier parts of the manuscript.

7. The conclusion should analyse the possible contribution of the assessed methods in the possible implementation in clinical practice.

Comments on the Quality of English Language

The work should be revised by an English-Native speaker.

Author Response

Thank you very much for taking the time to review this manuscript. Please find the detailed responses below and the corresponding revisions in the re-submitted files.

Response 1: Thank you for pointing this out. We agree with this comment. In the introduction, we presented the context of various hypotheses regarding PD pathogenesis. Mention exactly where in the revised manuscript this change can be found – page 2, paragraph 1, and line 4.

Response 2: We agree with this comment. We have described in the context of atypical parkinsonisms e.g. Corticobasal Syndrome. We have added the references you provided. Mention exactly where in the revised manuscript this change can be found–page 2, paragraph 2, and line 7.

Response 3: In the material section, we have included the history of transient ischemic attack as an exclusion criterium. Mention exactly where in the revised manuscript this change can be found–page 3, paragraph 3, and line 7.

Response 4: We have presented the clinical manifestation about the regions indicated in the classifier performance assessment. Mention exactly where in the revised manuscript this change can be found–page 8, paragraph 4, and line 1.

Response 5: We agree with this comment. In the limitation section, we have included the relatively wide age range as part of the limitations. Mention exactly where in the revised manuscript this change can be found–page 9, paragraph 2, and line 2.

Response 6:In the discussion section, we also made comparisons with related studies of SPECT. Mention exactly where in the revised manuscript this change can be found–page 8, paragraph 2, and line 4.

Response 7: In the conclusion, we pointed out that: these characteristic brain regions could become potential imaging markers of CBF to distinguish TD from PIGD. our findings provide an imaging basis for research on the neuropathological mechanism and personalized treatment, thereby optimizing clinical diagnostic and treatment approaches. Mention exactly where in the revised manuscript this change can be found–page 9, paragraph 3, and line 6.

Response to Comments on the Quality of English Language

As for the English revisions, we have used editing services and improved the level of English in the process of revision.

Reviewer 4 Report

Comments and Suggestions for Authors

The authors present an interesting study exploring the possibility of finding criteria unique to certain classifications of Parkinson’s disease, and utilising this to create an opportunity where this classification could be automatically determined based on imaging data. Briefly, the authors recruited 10-11 patients with diagnosed Parkinson’s disease and which represented one of two classifications of such. Utilising control healthy individuals, MRI imaging was used to determine unique patterns across the three populations, before this data was used in machine learning processes to identify those regions which presented unique traits associated with one or the other classification. While the study is limited in its scope, the authors do determine certain regions that may hold promise in this regard.

Overall, this is a well written and balanced study which provides new data and insights into classification of Parkinson’s disease, but interprets the data taking heed of the limitations inherent of the study.

In reviewing the article, I made some observations. The following should be considered when preparing a suitable revision.

1.       In Figure 2, the data appears to have titles that may have been default created by the software used. These labels are not straightforward to interpret, and it might be best if the authors adjust the wording on these graphs so it is clear as to what data set refers to which region of the brain.

2.       In figure 3, are the images produced based on data which was input from this particular study, or are the images ‘superficial’ in nature and simply included to aid the reader in identifying where in the brain the data pertains to? Either way, this should be made clearer so the image is interpreted appropriately.

Author Response

Thank you very much for taking the time to review this manuscript. Please find the detailed responses below and the corresponding revisions in the re-submitted files.

Response 1: Thank you for pointing this out. We agree with this comment. Therefore, we have adjust the wording on these graphs. The relevant labels are explained under the graphs, clearly showing what data set refers to which region of the brain.

Response 2: According to the sensitive brain regions screened by SVM, we input four related sensitive brain regions into BrainNetViewer for visualization, and displayed the related brain regions intuitively to aid the reader in identifying where in the brain the data pertains to.

Round 2

Reviewer 1 Report

Comments and Suggestions for Authors

I thank the authors for addressing my queries However please see below and help clarify 

1)Please correct me if I understood things wrongly. IN RESPONSE 1 authors have added in revised abstract “The length of the feature vector depends on the number of voxel values in each brain region”. So does the feature vector size change for each brain region for every subject?

I am not sure how can you train a classifier with each subject having different number of voxels in each brain region as feature vector (as I understand from authors reply). And then in testing phase again for each subject the number of features will vary across each brain region?

Adding to confusion Fig.1 shows voxel values of 170 brain regions (Voxel 1, Voxel 2…..Voxel 170) so does that denote that all voxel in each of the 170 regions are summed up to get Voxel 1,Voxel 2 to Voxel 170?

 2) Response 4 and their corresponding additions in the paper seems misleading. In Response 4 authors say that “Resting-state functional MRI is widely utilized in the study of PD motor subtypes [17-20] while there are few studies on ASL-MRI. It is necessary to consider an imaging marker of CBF to distinguish TD from PIGD”.

 Please add the rational or motivation why imaging marker of CBF is best suited to distinguish TD from PIGD. Is there a previous reference or what is the science behind this statement?

 3)In response 8 authors say “SVM can handle unbalanced data by adjusting the regularization parameter C to ensure that the model is not biased toward the dominant category”

Please mention in paper how adjustment of regularization parameter C was done?

Comments on the Quality of English Language

Minor editing and review of grammar needed

Author Response

Thank you very much for taking the time to review this manuscript. Please find the detailed responses below and the corresponding revisions in the re-submitted files.

Response 1: 

The feature vector size changes for each brain region for every subject.

We trained the model for the same brain region of all subjects in each experiment to test the diagnostic effect under the current brain region. According to the classification standard of AAL3 brain region template, a total of 170 brain regions were shown. Therefore, experiments were conducted for all 170 brain regions. In the experiment, each subject's current brain region would be used as the test set in turns due to adopting LOOCV. For example, when we targeted the Thal_VA_L in the AAL3 template for the experiment, the voxel value of Thal_VA_L for each subject was taken as a sample. When we performed experiments on other brain regions, since the number of voxel values in each brain region was different, the length of the feature vector in each experiment depended on the number of voxel values in each brain region. Mention exactly where in the revised manuscript this change can be found – page 4, paragraph 6, and line 5.

Figure 1 shows the voxel values of 170 brain regions, representing the voxel vectors of each region, rather than a single value obtained by summing all the voxels in each region. We also adjusted the confusing part of the Figure 1.

Response 2: In recent years, neuroimaging studies have shown that cerebrovascular lesions are common in Parkinson's disease patients. Therefore, PD is considered a disease related to abnormal cerebrovascular function . Mention exactly where in the revised manuscript this change can be found – page 2, paragraph 2, and line 4.

Previous studies have confirmed that dopaminergic neurons are attached to brain microvessels and cerebral blood flow (CBF) changes due to metabolic reduction caused by neuronal degeneration and death . However, this basic pathological change reflected in cerebral blood perfusion in patients with different motor subtypes of PD has not been confirmed by definite studies. Mention exactly where in the revised manuscript this change can be found – page 2, paragraph 2, and line 11.

Previous reference has shown that the PIGD group had a more predominantly posterior pattern of hypoperfusion and indeed basal ganglia hyperperfusion than the more temporo-parieto-frontal hypoperfusion of the TD group (which did not show areas of hyperperfusion). To our knowledge, the PD subtypes differences revealed in specific brain regions of CBF have not been previously investigated. Mention exactly where in the revised manuscript this change can be found – page 2, paragraph 3, and line 9.

Response 3: In the conducted experiment, a range of regularization parameter C values from 1 to 1000 were explored. Based on the obtained experimental results, it was determined that the current value of C exhibited optimal efficacy. Mention exactly where in the revised manuscript this change can be found – page 8, paragraph 2, and line 17.

Response to Comments on the Quality of English Language

As for the English revisions, We have made minor editing and reviewed of grammar. 

Reviewer 3 Report

Comments and Suggestions for Authors

I do not have any further comments.

Author Response

Thank you very much for taking the time to review this manuscript.